# An Under-Recognized Disease: A Rare Case of Idiopathic CD4 Lymphopenia Mislabeled as Primary Ciliary Dyskinesia

**DOI:** 10.3390/children9101534

**Published:** 2022-10-07

**Authors:** Fatimah Bukhamseen, Abdullah Al-Shamrani

**Affiliations:** 1Department of Pediatrics, King Fahd Hospital of the University, College of Medicine, Imam Abdulrahman Bin Faisal University, Dammam 34224, Saudi Arabia; 2Department of Pediatrics, Prince Sultan Military Medical City, Alfaisal University, Riyadh 12714, Saudi Arabia

**Keywords:** lymphopenia, bronchiectasis, ciliary dysfunction, immunodeficiency, case report

## Abstract

Idiopathic CD4 Lymphopenia is a heterogeneous condition, recognized in the late 20th century, with a wide spectrum of presentations, requiring a high index of suspicion to avoid misdiagnosing the condition. This case highlights the diversity in its clinical presentations in the context of an autosomal dominant pattern of inheritance. We are reporting a case of a nine-year-old child, initially labelled by her primary treating hospital as primary ciliary dyskinesia after presenting with chronic cough, purulent nasal discharge, and recurrent chest infections. She was referred to our facility, a tertiary center, as her condition marginally improved. After the patient has undergone a comprehensive diagnostic workup, including a gene study, she was found to be carrying a mutation known to cause idiopathic CD4 lymphopenia. Extended work up of her family showed that two of her siblings have inherited an autosomal dominant mutation from their mother who had a milder form of the disease. This condition is an extremely rare condition in children, which can be easily mislabeled. Thus, healthcare providers should avoid labeling certain long-standing diseases unless the diagnosis has been established. We encourage leveraging the use of the latest revolutionary genetic testing techniques to confirm the diagnosis of such puzzling cases.

## 1. Introduction

Idiopathic CD4 Lymphopenia (ICL) is a rare acquired immunodeficiency syndrome that has not been attributed to any transmissible infectious agents [1,2]. It was first recognized as a disease entity by the Centers for Disease Control and Prevention (CDC) in 1992 after several cases of human immunodeficiency virus (HIV)-negative with acquired immunodeficiency syndrome (AIDS)-like presentation were reported, with all other secondary causes of immunodeficiency being excluded. Many cases have been reported since then, revealing a heterogenous pattern to this condition, which encompasses a wide spectrum of diseases including asymptomatic immunological defect, opportunistic infections, neoplastic, and autoimmune conditions [3]. 

Recently, literature regarding the etiology and pathogenesis of ICL has been expanding, yet many questions about the condition remain unanswered. In efforts to shed light on possible inherited genetic etiological factors to this condition, we present a case of a nine-year-old girl presenting with recurrent chest infections and sinusitis, who was initially suspected to have primary ciliary dyskinesia. Genetic testing confirmed a rare mutation in Uncoordinated 119 (UNC119) gene, which she and her siblings have inherited from their mother; and thus was labeled as ICL.

## 2. Case Illustration

### 2.1. Clinical Presentation

A nine-year-old girl was referred to our facility, a tertiary center, as a case of recurrent chest infections, gastroesophageal reflux disease (GERD), and bronchial asthma. Upon presentation, she was complaining of frequent productive cough, occasionally yellowish, not seasonal, more in the first third of the night and not responding to bronchodilators. Her cough was associated with dyspnea, purulent nasal discharge, and nasal congestion. Her first chest infection was diagnosed at the age of one month, for which she was managed as a case of bronchiolitis. This episode was followed by several other admissions with similar respiratory complaints. As her cough persisted, she was started at the age of three years on a trial of fluticasone (50 mcg two puffs twice a day), montelukast granules (4 mg daily), and salbutamol (0.1 mg four puffs as needed). After the patient was started on asthma therapy trial, she exhibited only partial improvement as her exacerbation’s frequency reduced. At the age of five, she was diagnosed with GERD based on a barium swallow test and started on a proton pump inhibitor showing minimal improvement. Consequently, her primary treating hospital labeled her by the age of six based on her clinical features as primary ciliary dyskinesia (PCD).

Her family history (Figure 1) was positive for asthma-like symptoms experienced by her mother during childhood, and recurrent sinusitis in her three-year-old brother. Furthermore, her 15-year-old sister had two episodes of skin abscesses, one of which required surgical drainage. She also has a history of hearing difficulties, for which she is currently following up with an otolaryngologist to confirm the diagnosis of chronic otitis media. Other parts of the history, including surgical, perinatal, developmental history were all negative.

On examination, she was slightly tachycardic and in mild respiratory distress with a respiratory rate of 33 breaths per minute and desaturation of 94% at room air. Failure to thrive was evident as the patient was underweight of 17 kg at the third percentile. She had a positive Bacillus Calmette–Guérin vaccine scar, and grade 1 finger clubbing. Her chest auscultation revealed equal bilateral bronchovesicular breath sounds, with diffused crackles, most prominently on the right side. Otherwise, the systemic physical examination was unremarkable.

### 2.2. Diagnostic Workup

The patient condition necessitated a multidisciplinary approach, involving several sub-specialties, and requiring multiple laboratory workup and diagnostic procedures. Her initial workup, constituting of full blood count, electrolyte, and blood gases, were all within normal limits (Table 1). Similar to the findings of her primary treating hospital, her chest x-ray showed patchy infiltrates, primarily over the lower zones (Figure 2). It was consistent with the results of the computed tomography (CT) scans of the chest, which illustrated the presence of bronchiectasis involving the lower zones (Figure 3). Moreover, she underwent bronchoscopy and bronchoalveolar lavage a few days after starting antibiotics. Bronchoscopy findings showed mild erythematous change of the airway with no purulent secretions distally, while the lavage showed no growth of any bacterial organisms, including acid-fast bacilli. Although endoscopy failed to demonstrate any anatomical abnormality, severe reflux was noted on the barium swallow test (Figure 4). To assess the copious nasal discharge, the patient underwent a CT sinus scan and nasopharyngeal scope, confirming the diagnosis of infectious ethmoid and maxillary sinusitis on a background of allergic rhinitis. Immunological workup was carried out to rule out any primary immunodeficiency conditions, including lymphocyte subsets, immunoglobulin subclasses, tetanus, and diphtheria antibodies. All tests, except the lymphocyte subsets, demonstrated a normal result (Table 1). The standard spirometry showed mild partially reversible obstructive airflow limitation with FVC 75%, FEV1 of 65%, FEV1/FVC 71% and mid-expiratory airflow of 45% with 20% improvement post-challenge with bronchodilator.

In order to screen the diagnosis of PCD, nasal nitric oxide (NO) measurements were taken and were found to be normal. Ultrastructural evaluation with high-speed video microscopy (HSVM) was not conducted as it is unavailable in our center. Therefore, the genetic team was involved to exclude ciliopathy disorders and cystic fibrosis by performing Whole Exome Sequencing (WES) for the patient and her family members. Unexpectedly, the results came back positive for a heterogenous missense pathogenic variant of UNC119 gene (OMIMr: 604011, variant c.677T > C p. Leu226Pro). Extended workup for her family confirmed the same mutation in the mother and two of her siblings. Pathogenic variants in the gene are possibly causative for autosomal dominant immunodeficiency type 13, also known as idiopathic CD4 lymphopenia (ICL). It is a rare and heterogeneous syndrome defined by a reproducible reduction in the CD4 T-lymphocyte count (less than 300 cells per microliter or less than 20% of total T cells) in the absence of HIV infection or other known causes of immunodeficiency.

### 2.3. Therapeutic Interventions

The patient’s management plan for her respiratory condition consisted of antibiotics (ceftriaxone and azithromycin), airway clearance therapies to reduce mucus viscosity, facilitating its removal (bronchodilators hypertonic saline and dornase alfa), and aggressive chest physiotherapy. Additionally, inhaled nasal corticosteroid was used to treat the sinusitis alongside the antibiotics. For her severe reflux, the patient was started on high-dose esomeprazole for two weeks and then reduced to a standard dose. To support her growth and optimize her nourishment, Pediasure, a balanced supplemental nutrition, was added to her regular diet. 

The patient was discharged in a good condition with regular follow up with several experts from her multidisciplinary team. Her discharge medications consisted of asthma therapy, as per Global Initiative for Asthma guidelines, consisting of fluticasone (125 mcg twice a day) and montelukast chewable (5 mg), and salbutamol metered dose inhaler (0.1 mg four puffs) as needed. In addition, she was prescribed dornase alfa (2.5 mg once a day) and 3% hypertonic saline, and antibiotic in case of a pulmonary exacerbation. Counseling to the family was carried out in detail, and the possible potential use of immunological therapy in the future has been discussed, including intravenous immunoglobulin therapy and new biological agents.

## 3. Discussion

ICL was defined by the CDC as a reproducible reduction in CD4+ T-Lymphocyte, with counts less than 300 cells per microliter or less than 20% of total T cells, which is not attributed to an identifiable cause, including viral infections, primary immunodeficiencies, or any therapy associated with lymphopenia [4]. Due to its rarity, most of what is known about the disease is concluded from scattered cases reported in the literature [1]. Thus, the disease remains poorly understood regardless of the expanding research concerning its pathogenesis.

Many hypotheses regarding the underlying pathology leading to CD4+ T cell depletion have been proposed. As initial cases have been described in HIV-negative individuals, a viral cause was initially suspected; however, no definite viral causative agent was as yet identified [1,5]. An increase in the apoptotic activity of CD4+ T cells in ICL patients was observed, which was proposed to be secondary to impaired Interleukin (IL)-2 and IL-7 regulation or increased Fas-induced apoptosis. In addition, several genetic mutations have been linked to the depletion of CD4+ count, suggesting a genetic predisposition to the disease. Uncoordinated 119 protein is a signaling adaptor protein that is responsible for lymphocyte-specific kinase (Lck) activation. Lck is a key T cell tyrosine kinase, that initiates signaling from T cell receptor (TCR) [6]. Its critical role in T cells development and activation is well established [1,3]. A mutation affecting UNC119 gene has been recognized as a precursor to the development of ICL, as it has been identified in several cases, including ours [1,6]. Lck and Unc119 mutations lead to severe combined immune deficiency phenotypes [7]. Other mutations implicated in the pathogenesis of ICL include; serine threonine kinase 4, magnesium transporter gene, and recombination activating gene1 (RAG1) mutations [1,6]. The heterozygous mutation of RAG1, which affects the TCR repertoire as it is responsible for TCR’s beta chain [1,8], was suggested to account for alteration of the disease course, as it has been linked to milder phenotypes of the disease [3]. This could explain the vast difference in the clinical picture among the affected family members reported in this case, with the mother and brother having a more subtle manifestation compared to the patient and her sister, with tendency to improve with time. 

Due to the heterogeneous nature of this condition, the clinical presentation reported in the literature is vastly diverse. It ranges from asymptomatic immunological defect to severe fatal opportunistic infections and neoplasms. Low CD4+ count increases the risk of both opportunistic and non-opportunistic infections [1,2], with cryptococcus and nontuberculous mycobacterium infections being the most common presentation in ICL patients [1,2,3]. A wide spectrum of infections is described in the literature, among which skin, neurological, and respiratory involvement were the most common [1]. Consistent with the described literature, the affected family members in our case had primarily respiratory manifestations, except for her sister who presented with recurrent skin infections. Other manifestations include autoimmune and neoplastic conditions [1,2,3]. Similar to our patient, Kose et al. have described a case of a 12-year-old boy, who presented with bronchiectasis and clubbing; however, the boy also had candidiasis and hypogammaglobinemia A [9]. Several cases of ICL presenting with bronchiectasis have been described [6,10]. This could be attributed to the patients’ susceptibility to more recurrent and severe respiratory infections, leading to persistent inflammation of the airways and structural lung damage. In addition to these findings, our patient also suffered from sinusitis and failure to thrive, which raised the suspicion of ciliary dysfunction [11]. Another unique manifestation in our patient is the severe reflux disease. However, there is no data in the literature that suggests an association between GERD and ICL.

In this paper, we present a case of a nine-year-old girl with recurrent infections, found to have inherited UNC119 gene mutation from her mother, who was initially labeled based on her clinical features as PCD. However, when nasal NO measurement was carried out to confirm the diagnosis, it was found to be normal, which made the diagnosis unlikely [12]. Normal or high nNO still account for less than 5% of known PCD [13]. According to the European Respiratory Society guidelines for the diagnosis of PCD, a high or normal nasal NO plus normal HSVM excludes the diagnosis [14]. Although HSVM was not carried out due to its unavailability in out center, the American Thoracic Society clinical practice guidelines for the diagnosis of PCD recommends using an extended genetic testing panel over a standard genetic panel and HSVM. This applies in particular to cases with normal nasal NO, as most PCD forms with a normal nasal NO tend to have a normal or nondiagnostic ultrastructural evaluation with HSVM [15]. Our patient did not show any signs of respiratory distress in the neonatal period, as would the majority of those with PCD, which could be as mild as transient tachypnea of newborns or severe enough to require positive pressure support. Furthermore, there was no recurrent otitis media in the first two years, nor speech delay pointing to conductive hearing loss, and there was no dextrocardia or situs inversus. In contrast, the wet cough and purulent nasal discharge were consistent with chronic sinus infections, and the mild bronchiectasis and repeated recurrent chest infection were very suggestive of PCD. [16] The dornase alfa improve mucociliary in non-CF or idiopathic bronchiectasis including primary ciliary dyskinesia, pneumonia, and severe asthma, and some ventilated cases in PICU [17,18]. Skin manifestation was negative in our patient but was positive in her sister, however it a common manifestation of ICL in 52% of the cases, as reported by Vijayakumar S et al. [1,19,20]. 

Meanwhile, our patient fulfills the criteria of ICL as her CD4+ count was found to be less than 300, although her total white blood cell count was normal. Although her repeated chest infections and bronchiectasis could be justified partially by GERD, no other explanation for her recurrent sinusitis and CD4+ depletion was identified. WES was conducted for the patient and her family, revealing the same mutation occurring in the mother and her three children, confirming an autosomal dominant inheritance pattern. Other cases of ICL in siblings have been reported, suggesting a familial predisposing factor to the development of the disease [10]. Nonetheless, we did not come across any case reports that have identified the same gene among family members. 

The management of ICL is mainly focused on the treatment of the presenting illness. Patients with autoimmune diseases are generally managed based on disease-specific guidelines that have been established. Similar to HIV-positive patients, infections screening and prophylaxis in ICL patients are guided by CD4 counts [1]. Vaccine recommendations are mainly focused on minimizing the risk of severe pneumonia by administering the polysaccharide Pneumoccocal and the annual Influenza vaccine [1,20]. Moreover, as recent evidence has been published regarding IL-2 and IL-7 dysregulation role in the disease pathogenesis, several investigators have reported IL-7 and IL-2 therapy to be successful, resulting in infections clearance and higher CD4 counts [1,3,5].

## 4. Conclusions

ICL is an extremely rare condition in children, which can be overlooked easily due to its heterogenicity. To avoid misdiagnosing ICL, detailed history, meticulous examination, and a multidisciplinary approach are strongly recommended. Furthermore, health care providers should exercise caution by making the necessary investigations to confirm the diagnosis prior to labeling long-standing diseases such as clinical primary ciliary dyskinesia. Hence, we encourage leveraging the use of the latest revolutionary genetic testing techniques such as WES, especially in the presence of alarming clinical features including but not limited to chronic cough, repeated chest infections, purulent nasal discharge, clubbing, or cystic bronchiectasis.

## Figures and Tables

**Figure 1 children-09-01534-f001:**
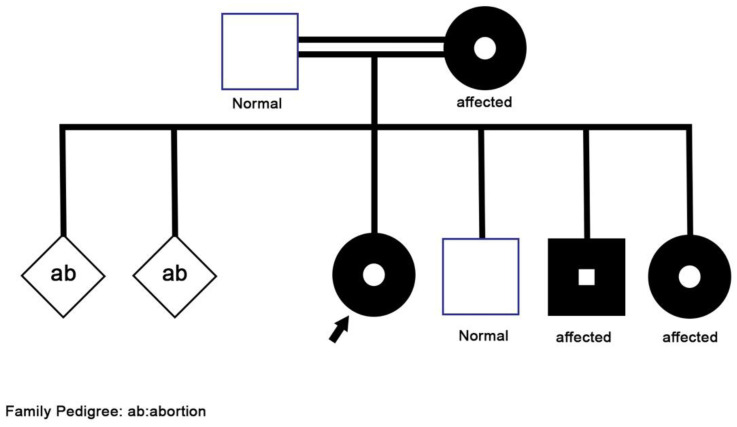
Family Pedigree.

**Figure 2 children-09-01534-f002:**
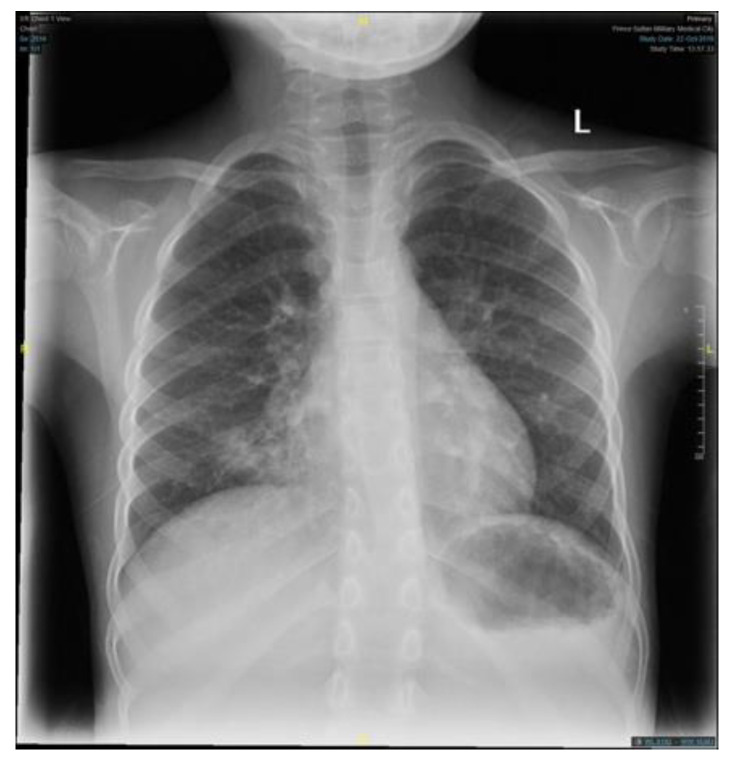
A plain chest X-ray, an anteroposterior film, showing a radioopaque opacity infiltrate seen over the right and left lower zone with mild hyperinflation. (L = Left).

**Figure 3 children-09-01534-f003:**
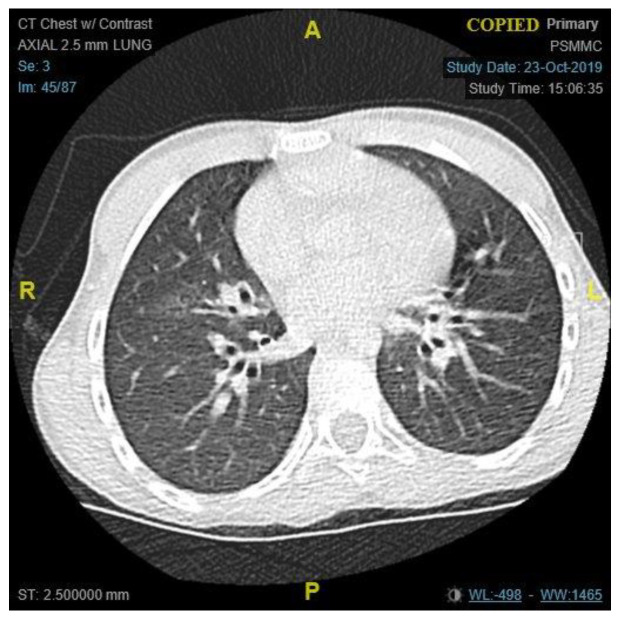
A chest CT scan, an axial view of lung window, showing cystic changes strongly consistent with bronchiectasis in lower lobes. (A = Anterior, P = Posterior, R = Right, L = Left).

**Figure 4 children-09-01534-f004:**
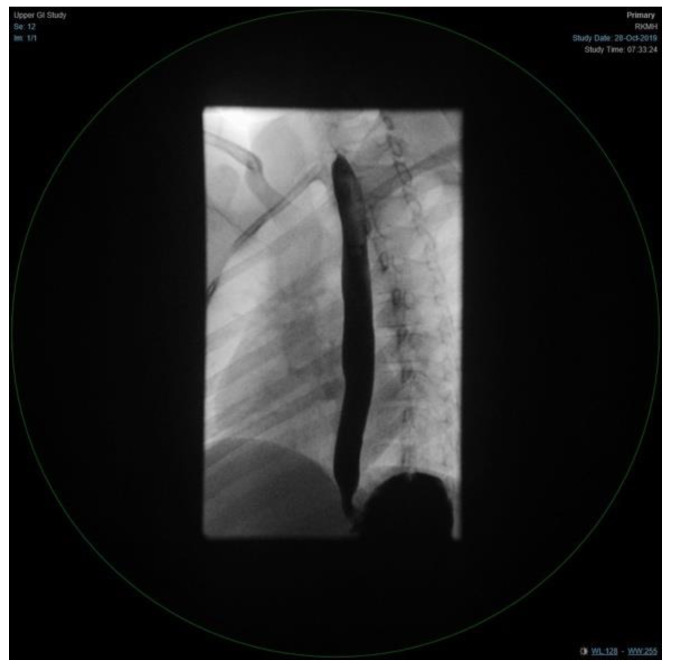
An upper gastrointestinal study, consistent with severe reflux.

**Table 1 children-09-01534-t001:** Laboratory Investigations.

Test	Result
White Blood Cells	7.5 × 10^9^ cell/L
Hemoglobin	10.5 g/dL
Platelet	481 × 10^9^ cell/L
Absolute Neutrophils count	3 × 10^9^ cell/L
Absolute Lymphocyte count	3.3 × 10^9^ cell/L
Absolute Monocyte count	0.8 × 10^9^ cell/L
Absolute Basophil count	0 cell/L
Eosinophils	0.4 × 10^9^ cell/L
Total Helper T-Cells (CD4+) % (normal value)	19.4% (31–47)
Total Helper T-Cells (CD4+) number	290 cell/L
IgG level	12.2 g/L
IgM level	0.88 g/L
Total IgA level	1.6 g/L
IgE level	>18.8 IU/mL
Diphtheria Antitoxoid Antibodies	0.16
Tetanus Antitoxoid IgG Antibodies	0.21

## Data Availability

Not applicable.

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
