# Peer review of "An Under-Recognized Disease: A Rare Case of Idiopathic CD4 Lymphopenia Mislabeled as Primary Ciliary Dyskinesia"

_children, 2022, doi:10.3390/children9101534_

Round 1

Reviewer 1 Report

The host defenses of the lung depend on the complex and integrated functions of mucociliary clearance (MCC) with cellular and antibody-mediated immune responses. Not surprising, since immunological defects can have features that overlap with primary ciliary dyskinesia, including chronic suppurative infections involving the lungs, middle ear, and paranasal sinuses. In this interesting manuscript, Drs. Bukhamseen and Al-Shamrani report a child with idiopathic CD4 Lymphopenia who was misdiagnosed with primary ciliary dyskinesia. 

While their observations are not novel, and the paper has several weaknesses that should be addressed. Their conclusion, however, “health care providers should exercise caution by taking the necessary investigations to confirm the diagnosis prior to labeling long-standing diseases such as clinical primary ciliary dyskinesia” is an important message. 

The subject’s clinical history is lacking, but based on the information provided, she should have never been diagnosed with primary ciliary dyskinesia in the first place.

For instance, the subject’s sister has a history of recurrent suppurative skin infections, a feature not associated with primary ciliary dyskinesia. Thus, primary immunodeficiency should be considered more likely.

The diagnosis of primary ciliary dyskinesia requires classic phenotypic features in conjunction with diagnostic testing [Leigh MW, et al. Ann Am Thorac Soc. 2016;13:1305-13]. Did this girl have features consistent with the diseaseMore details regarding her early clinical presentation are needed. Based on her imaging studies, she did not have a thoracic laterality defect, but was she born at term and have neonatal respiratory distress?  Did she have a “wet” cough or non-seasonal nasal discharge every day during early infancy and childhood? The authors commented that she has had “hearing difficulties,” but did she have persistent middle ear effusions?

The authors stated that “in order to confirm the diagnosis of primary ciliary dyskinesia, nasal nitric oxide (NO) measurements were done,” which is not standard of care.  While nasal nitric oxide can be sensitive and specific in cooperative children and adults with primary ciliary dyskinesia, reduced nasal nitric oxide concentrations alone are never sufficient to “confirm” primary ciliary dyskinesia. And normal nasal nitric oxide concentrations do not necessarily exclude the diagnosis.

Though not cited, the American Thoracic Society clinical practice guidelines was modified, recommending immunological testing for people who have features suggestive of primary ciliary dyskinesia but normal nasal nitric oxide levels [Shapiro, et alAm J Respir Crit Care Med. 2020;202:476-7].

Figure 4 was not available for review.

Regarding the gene, it’s stated in the paper that “Pathogenic variants in the gene are possibly causative for autosomal dominant immunodeficiency type 13, also known as idiopathic CD4 lymphopenia (ICL).” Is there still uncertainly whether mutations in this gene cause disease?  Has this variant been described in the literature as “pathogenic.” It was not cited in the references. If not, how did they determine that it is disease-causing?

Since the authors believe she has bronchiectasis secondary to primary immunodeficiency, idiopathic CD4 lymphopenia, why was she treated for asthma (inhaled fluticasone, montelukast, etc). Was spirometry performed? Does she have clinical evidence of airway hyperresponsiveness or hyperreactivity?

I’m also curious why the authors chose to treat her with human recombinant DNAse, since a large randomized clinical trial in adults with non-cystic fibrosis bronchiectasis, including subjects with PCD, showed worsening lung function and increased frequency of exacerbations [O'Donnell AE, et alrhDNase Study Group. Chest. 1998;113:1329-34]. 

Reviewer 2 Report

This work brings a case report of rare disease - Idiopathic CD4 lymphopenia. 

Some comments for authors:

Change the name of the paper - try to exclude PCD from the title. The clinical manifestation in this child were not typical for PCD phenotype at this age and also not all diagnostic methods for PCD were used. It could lead to misinterpretation that the PCD and idiopathic CD4 lymphopenia can be similar group of diseases. 

Author Response

We thank the reviewer for this great paper that was suggested.

Agreed with the reviewer that the case is not typical PCD.

Agreed we do not have the whole diagnostic facility as HSVM

The PCD was Initially described by Kartagener et al 1936, until the genetic aspect while ICL which was recognized in 1992 still poorly understood and presentation is diverse and the evidence in literatures still restricted to case reports

Still in non-expert hand and due to the diversity of the ICL might be confused with PCD especially if the appropriate work up for PCD not completed.

 We would like the reviewer to allow us keeping the title as it is, this a real fact in our case (when referred to us as clinical PCD for confirmation).